# *ALK*, *ROS1*, *RET* and *NTRK1–3* Gene Fusions in Colorectal and Non-Colorectal Microsatellite-Unstable Cancers

**DOI:** 10.3390/ijms241713610

**Published:** 2023-09-02

**Authors:** Rimma S. Mulkidjan, Evgeniya S. Saitova, Elena V. Preobrazhenskaya, Karimat A. Asadulaeva, Mikhail G. Bubnov, Ekaterina A. Otradnova, Darya M. Terina, Sofia S. Shulga, Darya E. Martynenko, Maria V. Semina, Evgeniya V. Belogubova, Vladislav I. Tiurin, Priscilla S. Amankwah, Aleksandr S. Martianov, Evgeny N. Imyanitov

**Affiliations:** 1Department of Tumor Growth Biology, N.N. Petrov Institute of Oncology, 197758 St. Petersburg, Russia; mulkidzhan17@gmail.com (R.S.M.);; 2Department of Medical Genetics, St. Petersburg Pediatric Medical University, 194100 St. Petersburg, Russia

**Keywords:** microsatellite instability, gene rearrangements, PCR, NGS, colorectal cancer, unbalanced expression, *ALK*, *RET*, *NTRK1*, *NTRK2*, *NTRK3*

## Abstract

This study aimed to conduct a comprehensive analysis of actionable gene rearrangements in tumors with microsatellite instability (MSI). The detection of translocations involved tests for 5′/3′-end expression imbalance, variant-specific PCR and RNA-based next generation sequencing (NGS). Gene fusions were detected in 58/471 (12.3%) colorectal carcinomas (CRCs), 4/69 (5.8%) gastric cancers (GCs) and 3/65 (4.6%) endometrial cancers (ECs) (*ALK*: 8; *RET*: 12; *NTRK1*: 24; *NTRK2*: 2; *NTRK3*: 19), while none of these alterations were observed in five cervical carcinomas (CCs), four pancreatic cancers (PanCs), three cholangiocarcinomas (ChCs) and two ovarian cancers (OCs). The highest frequency of gene rearrangements was seen in *KRAS/NRAS/BRAF* wild-type colorectal carcinomas (53/204 (26%)). Surprisingly, as many as 5/267 (1.9%) *KRAS/NRAS/BRAF*-mutated CRCs also carried tyrosine kinase fusions. Droplet digital PCR (ddPCR) analysis of the fraction of *KRAS/NRAS/BRAF* mutated gene copies in kinase-rearranged tumors indicated that there was simultaneous co-occurrence of two activating events in cancer cells, but not genetic mosaicism. CRC patients aged above 50 years had a strikingly higher frequency of translocations as compared to younger subjects (56/365 (15.3%) vs. 2/106 (1.9%), *p* = 0.002), and this difference was particularly pronounced for tumors with normal *KRAS/NRAS/BRAF* status (52/150 (34.7%) vs. 1/54 (1.9%), *p* = 0.001). There were no instances of MSI in 56 non-colorectal tumors carrying *ALK*, *ROS1*, *RET* or *NTRK1* rearrangements. An analysis of tyrosine kinase gene translocations is particularly feasible in *KRAS/NRAS/BRAF* wild-type microsatellite-unstable CRCs, although other categories of tumors with MSI also demonstrate moderate occurrence of these events.

## 1. Introduction

Microsatellite instability (MSI), being a consequence of deficient mismatch repair (dMMR), is manifested by multiple mutations affecting repetitive DNA sequences [1,2]. MSI may occur in tumors associated with Lynch hereditary cancer syndrome. These malignancies arise in subjects with inherited pathogenic variants in the *MLH1*, *MSH2*, *MSH6*, PMS2 or *EPCAM* genes, and their development involves a somatic second-hit inactivation of the involved member of the dMMR pathway [3]. MSI is also characteristic of some sporadic malignancies, being attributed to *MLH1* promoter hypermethylation [4]. Microsatellite-unstable carcinomas have an increased tumor mutation burden (TMB) and are responsive to inhibitors of immune checkpoints [5,6].

MSI is particularly common for colorectal carcinomas, with approximately 5–15% of tumors displaying this phenotype. Microsatellite-unstable CRCs often carry mutations leading to MAPK signaling pathway activation, particularly amino acid substitutions in the *KRAS*, *NRAS* and *BRAF* genes [4,7]. Somewhat unexpectedly, MSI-CRCs were repeatedly shown to contain rearrangements in genes encoding receptor tyrosine kinases [2,8,9,10]. A recent study revealed that these translocations are related to an increased frequency of mutations within the G:C-reach intronic regions of the involved genes [11]. There are several drugs targeting ALK, ROS1, RET and NTRK1–3 tyrosine kinases; therefore, the detection of these gene fusions is of high clinical importance [10,12]. In addition to CRC, MSI is characteristic of several non-colorectal cancer types, particularly gastric and endometrial carcinomas [13,14]. It has not yet been systematically studied whether tyrosine kinase gene rearrangements occur at noticeable frequencies in microsatellite-unstable cancers arising in organs other than the colon.

A comprehensive analysis of *ALK*, *ROS1*, *RET* and *NTRK1–3* gene rearrangements requires RNA-based next generation sequencing, which is an expensive technique [2]. We have developed an efficient laboratory screening procedure for *ALK*, *ROS1*, *RET* and *NTRK1–3* translocations, which is largely based on an analysis of 5′/3′-end unbalanced expression of these genes [15,16]. When the gene is not affected, the number of transcripts corresponding to its kinase portion and to the upstream nucleotide sequences are equal. *ALK*, *ROS1*, *RET* and *NTRK1–3* rearrangements usually result in fusion of the kinase domain to the actively transcribed gene. In the latter case, the expression of the kinase-domain-related portion of the receptor tyrosine kinase is elevated as compared to sequences located upstream to the breakpoint. This pipeline, being coupled with the identification of the *ALK*, *ROS1*, *RET* and *NTRK1–3* fusion variants, allows for an analysis of a large number of tumor samples

## 2. Results

### 2.1. MSI Detection

MSI was detected in 743/14,111 (5.3%) CRCs, 84/1756 (4.8%) GCs, 91/506 (18%) ECs, 7/459 (1.5%) cervical carcinomas, 4/64 (6.3%) cholangiocarcinomas, 5/339 (1.5%) ovarian cancers and 4/474 (0.8%) pancreatic cancers (Appendix A). These frequencies are in good agreement with the published data [1]. Since we utilized both single-marker *BAT26* testing and the pentaplex panel (*BAT25*, *BAT26*, *NR21*, *NR22* and *NR24*), we evaluated whether the differences between these two methods are likely to influence the results. We considered the *BAT26* length in 471 CRCs, 69 GCs and 66 ECs that were microsatellite-unstable by the pentaplex assay. Strikingly, only 3 out of 586 of these tumors had normal *BAT26* status, indicating that the methodology of MSI testing had no significant influence on the results.

### 2.2. Analysis of Gene Rearrangements

An analysis of gene rearrangements was considered for 659 tumors carrying MSI (501 CRCs, 73 GCs, 69 ECs, 6 CCs, 4 ChCs, 4 PanCs and 2 OCs). There were 40/659 (6.1%) cases that failed to pass the RNA quality controls, as the referee gene in these samples emerged after the 35th PCR cycle (30/501 (6.0%) CRCs, 4/73 (5.5%) GCs, 4/69 (5.8%) ECs, 1/6 (16.7%) CCs and 1/4 (25%) ChCs). The remaining 619 MSI-positive samples (471 CRCs, 69 GCs, 65 ECs, 5 CCs, 4 PanCs, 3 ChCs and 2 OCs) were subjected to an analysis of gene fusions.

The 5′/3′-end expression imbalance for receptor tyrosine kinases was observed in 59 tumors (*ALK*: 8; *RET*: 8; *NTRK1*: 24; *NTRK2*: 2; *NTRK3*: 17). All these samples were available for variant-specific PCR, and the latter procedure resulted in the identification of rearrangements in 28/59 (47.5%) samples. Variant-specific PCR failed to reveal translocations in 31/59 (52.5%) tumors with the 5′/3′-end unbalanced expression. Only 13 of these tumors were analyzed by NGS, as the majority of samples were returned to primary hospitals after the completion of MSI testing and, therefore, were not available for the new round of RNA isolation. Strikingly, NGS revealed gene translocations in all these tumors (*SPTBN1::ALK (S7;A20)* (*n* = 2); *ETV6::NTRK2 (E5;N15)* (*n* = 2); *NCOA4::RET (N9;R12)* (*n* = 2) and one each *NCOA4::RET (N9del501;R12)*; *STRN::ALK (S5;ins53A20)*; *RBBP8::ALK (R8;A20)*; *TPR::NTRK1 (T15;N10)*; *TPR::NTRK1 (T20;N11)*; *ZKSCAN1::NTRK1 (Z3;N12)* and *LMNA::NTRK1 (L9ins44;del74N11)*) (Table 1). Taken together, the above data allow for the assumption that the majority of the remaining 18 tumors, which were not available for NGS, also carry rare tyrosine kinase gene rearrangements.

Our screening procedure involved simultaneous determination of 5′/3′-end unbalanced expression and the use of variant-specific PCR for common translocations. All tumors carrying fusions in the *ALK*, *NTRK1* and *NTRK2* kinases demonstrated expression imbalance. However, the 5′/3′-end expression had relatively poor performance for *RET* and *NTRK3* gene fusions, as 4/9 (44.4%) and 2/11 (18.2%) tumors with rearrangements of these genes, respectively, showed normal expression of these genes.

In addition to the above-mentioned well-known gene rearrangements, we performed an analysis of novel fusions (*BCR::PKHD1* and *CLIP1::LTK*) that were recently identified in lung carcinomas [17,18]. None of the microsatellite-unstable carcinomas carried these translocations.

### 2.3. Pattern of Gene Rearrangement in Various Categories of Microsatellite-Unstable Tumors

There were 47 microsatellite-unstable tumors with identified translocation variants (*ALK*: 6; *RET*: 9; *NTRK1*: 19; *NTRK2*: 2; *NTRK3*: 11) (Figure 1). In addition, there were 18 tumors with 5′/3′-end unbalanced expression (*ALK*: 2; *RET*: 3; *NTRK1*: 5; *NTRK3*: 8), which were highly likely to carry rearrangements in the above genes, but the involved translocation variants could not be identified due to the above-mentioned technical limitations. 

The frequency of gene fusions in microsatellite-unstable tumors approached 58/471 (12.3%) for CRC, 4/69 (5.8%) for GC and 3/65 (4.6%) for EC. The study included 204 CRCs negative for mutations in *KRAS*, *NRAS* or *BRAF* oncogenes; when considering samples with identified translocation variants, 42 (20.6%) of these tumors carried gene rearrangements (*ALK*: 6; *RET*: 9; *NTRK1*: 16; *NTRK2*: 1; *NTRK3*: 10). In addition, 11 *KRAS/NRAS/BRAF* mutation-negative CRCs demonstrated 5′/3′-end expression imbalances for one of the receptor tyrosine kinases (*ALK*: 2; *RET*: 3; *NTRK1*: 3; *NTRK3*: 3); however, the fusion variants could not be identified due to technical reasons. In total, 53/204 (26%) *KRAS/NRAS/BRAF* mutation-negative CRCs appeared to carry actionable gene fusions (*ALK*: 8; *RET*: 12; *NTRK1*: 19; *NTRK2*: 1; *NTRK3*: 13).

Somewhat surprisingly, as many as 5/267 (1.9%) CRCs with activating mutations in the *KRAS*, *NRAS* or *BRAF* genes were found to have kinase gene rearrangements (Figure 2, Table 1). In addition, the gastric tumor with *TPM3::NTRK1 (T8;N10)* rearrangement also simultaneously carried the p.G12C mutation in the *KRAS* oncogene. Activating genetic lesions in genes involved in the MAPK pathway are usually mutually exclusive; therefore, the coincident occurrence of gene fusions and *KRAS/NRAS/BRAF* mutations is intriguing. This coincidence may occur due to the presence of the above events in distinct cells, i.e., the mosaicism of activating genetic lesions, or due to the simultaneous occurrence of two mutations in the same cell. We evaluated the fraction of *KRAS/BRAF*-mutated cells in these tumors using digital droplet PCR (Appendix A). The obtained data were highly concordant with the results of the visual inspection of the slides, thus suggesting that *KRAS/BRAF* mutations are not mosaic but present in all tumor cells. It is noteworthy that all but one of the above-described tumors with tyrosine kinase rearrangements showed 5′/3′-end unbalanced expression of the affected gene; this imbalance could not be detected if only minor fractions of the tumor cells carried a fusion (Table 1). Furthermore, the only tumor with a non-altered expression pattern carried a translocation in the *NTRK3* gene; as mentioned above, alterations in this kinase are not always accompanied by changes in the 5′/3′-end transcript ratio.

A high number of microsatellite-unstable CRCs carrying gene rearrangements permitted the analysis of clinical correlations. Male/female ratios were similar in fusion-positive and fusion-negative tumors (Appendix A, Appendix A). Patients aged above 50 years had a strikingly higher frequency of translocations as compared to younger subjects (56/365 (15.3%) vs. 2/106 (1.9%), *p* = 0.002). This difference was even more pronounced for tumors with normal *KRAS/NRAS/BRAF* status (52/150 (34.7%) vs. 1/54 (1.9%), *p* = 0.001; Supplementary Appendix A). 

### 2.4. MSI Analysis in Non-Colorectal Tumors Carrying ALK/ROS/RET/NTRK Rearrangements

This and other studies have demonstrated that a subset of microsatellite-unstable tumors carries tyrosine kinase gene fusions. We questioned whether the same overlap between these two events is observed when the MSI testing is applied to non-colorectal kinase-rearranged carcinomas. We utilized pentaplex panel testing for 23 lung carcinomas, 23 sarcomas, 4 thyroid carcinomas, 3 salivary gland tumors, and 3 pancreatic carcinomas with oncogenic gene translocations (*ALK*: 31; *ROS1* 10; *RET*: 9; *NTRK1*: 2; *NTRK3*: 4). None of the above tumors demonstrated microsatellite alterations.

## 3. Discussion

This is apparently the largest single-center study to systematically evaluate gene rearrangements in microsatellite-unstable tumors belonging to various cancer types. It has been confirmed that *ALK*, *RET* and *NTRK1–3* gene rearrangements are very frequent in *KRAS/NRAS/BRAF* mutation-negative CRCs, especially in patients aged above 50 years, although these events may also occur in colorectal tumors carrying *RAS/RAF* mutations as well as in non-colorectal MSI-positive CRCs.

The results of this report are consistent with a recent study by Madison et al. [11], who demonstrated that the emergence of gene rearrangements in CRCs is attributed to the mutagenic effects of microbiota-derived butyrate and the consequent generation of 8-oxoguanine. These data convincingly explain the significant differences in the incidence of gene rearrangements between colorectal and non-colorectal malignancies. However, moderate frequencies of tyrosine kinase fusions were observed in gastric and endometrial tumors, suggesting that microsatellite-unstable cells may acquire translocations even in the absence of the influence of gut microbes.

Our report confirms that alterations in the genes belonging to the MAPK pathway are generally mutually exclusive. Indeed, activation of a single member of this molecular cascade, be it a receptor tyrosine kinase or *KRAS*, *NRAS* or *BRAF* oncogene, is usually sufficient to drive the signaling. Furthermore, the above-mentioned genetic alterations are usually considered to be more or less equivalent in terms of phenotypic consequences [7]. Interestingly, despite this apparent equivalence, the frequencies of alterations in particular genes vary considerably between MSI-positive and MSI-negative CRCs. Microsatellite-unstable CRCs have approximately a twice lower frequency of *KRAS* mutations but approximately a four times higher incidence of *BRAF* mutations as compared to MSI-negative tumors [7,19]. Furthermore, while kinase gene rearrangements are common in CRCs with MSI, they are exceptionally rare in MSI-negative colorectal carcinomas [11]. Overall, the cumulative frequency of the activation of the MAPK cascade is similar in CRCs with and without MSI; for example, 320/471 (67.9%) microsatellite-unstable CRCs analyzed in this study had evidence of genetic alteration of the MAPK pathway (Figure 2), which is very close to the estimates obtained for microsatellite-stable colorectal carcinomas [19]. The remaining 30–40% of CRCs do not have overt genetic alterations within this signaling cascade and deserve further investigations [7]. Interestingly, the analysis of our dataset revealed a few instances of the simultaneous occurrence of genetic events affecting two distinct oncogenes. Several prior investigations produced similar examples [19,20,21]. Single-cell sequencing may permit reliable discrimination between mutation mosaicism and the true co-occurrence of several MAPK activating events within the same cell [22]. Here, we utilized ddPCR for a rough analysis of the fraction of *KRAS/BRAF*-mutated cells, which strongly suggested that these mutations are truncal but not mosaic (Appendix A). Interestingly, there were two CRCs with simultaneous occurrences of *BRAF* p.V600E mutations and *NTRK* translocations. These tumors are unique with regard to clinical opportunities, as they have three highly actionable targets (MSI for immune therapy, *BRAF* p.V600E substitution for combined EGFR/BRAF inhibition and NTRK activation for the use of TRK inhibitors) [4,7,12].

Gene fusions occurred at high frequencies in CRC patients aged above 50 years, but were uncommon in younger subjects (Appendix A). This age threshold is commonly utilized for discrimination between sporadic CRCs, which develop microsatellite instability due to somatic hypermethylation of the *MLH1* promoter, and hereditary CRCs, which are attributed to the biallelic mutation-driven inactivation of DNA mismatch repair genes [3,4,7]. In this respect, tyrosine kinase translocations are similar to the *BRAF* p.V600E mutation, which is a validated marker for the exclusion of microsatellite-unstable CRCs from Lynch syndrome germline testing [3,7].

The spectrum of gene fusions in microsatellite-unstable tumors has some characteristic features. *NTRK1–3* gene fusions are relatively frequent in some pediatric tumors and sarcomas, although they are exceptionally rare in common cancer types [23,24,25]. However, *NTRK1–3* rearrangements compose the majority of the gene translocations observed in MSI-positive carcinomas. Similar to other tumor types [25], *TPM3::NTRK1*, *EML4::NTRK3* and *ETV6::NTRK3* fusion variants represented the majority of *NTRK1–3* translocations identified in this study. *ALK* and *RET* gene rearrangements are particularly frequent in lung carcinomas. While common translocation variants constitute over 90% of *ALK* fusions in pulmonary malignancies [26], only one such rearrangement was detected in our dataset (*EML4::ALK (E6;A20)*). Similarly, *KIF5B::RET* fusions represent more than 70% of *RET* alterations in lung cancers [16]; however, none of the *RET*-rearranged microsatellite-unstable tumors carried this variant. 

This study has some limitations. In particular, we did not have survival data for the patients; therefore, we could not evaluate whether the presence of gene rearrangements affects disease outcomes. MSI detection was based on a standard PCR protocol. Although this approach is reliable for the detection of tumors with highly unstable microsatellite repeats, it may miss a subset of mismatch repair-deficient carcinomas, which have low rates of cell proliferation and, therefore, low numbers of mutations in mononucleotide tracks [5,27]. The discordance between the immunohistochemical (IHC) evaluation of MMR proteins and the PCR analysis of MSI is mainly observed for tumors arising outside the gastrointestinal tract [5,27]. Therefore, given the preferential occurrence of gene fusions in CRCs, it is highly unlikely that these MMR-deficient but microsatellite-stable tumors would carry kinase-activating rearrangements. 

This study detected 31 samples with 5′/3′-end unbalanced expression, in which variant-specific PCR failed to identify known gene rearrangements. Only 13 of these tumors were available for NGS, and all these samples contained rare fusion variants. It is highly likely that the majority of the remaining 18 tumors (*ALK*: 2; *RET*: 3; *NTRK1*: 5; *NTRK3*: 8) also carry uncommon types of rearrangements in the mentioned genes. The poor availability of microsatellite-unstable tumor samples for RNA-based NGS is attributed to the study design. The standard procedure for sample processing in our laboratory involves the simultaneous isolation of DNA and RNA, followed by cDNA synthesis in the same tube. This protocol is not compatible with subsequent RNA-based NGS sequencing; therefore, we needed to retrieve the tissue samples and subject them to a new round of RNA isolation. This effort turned out to be inefficient, as the majority of archival blocks were returned to the primary hospital immediately after the completion of standard molecular testing. This described drawback can be easily resolved if the aliquot of the DNA/RNA sample is stored without subsequent cDNA synthesis and, therefore, used for NGS whenever necessary. We have now incorporated this amendment in the processing of those tumors that may potentially require NGS testing. It also has to be acknowledged that the administration of ALK and NTRK inhibitors is not necessarily based on the identification of particular translocation variants, as the FISH assay is an approved method for the analysis of these genes [28]. Overall, 5′/3′-end unbalanced expression very rarely produces false-positive results, i.e., virtually all tumors identified by this assay indeed carry gene rearrangement [15,16]. Therefore, in theory, the results of 5′/3′-end unbalanced expression *per se* may be sufficient to guide therapy in some circumstances.

## 4. Materials and Methods

The analysis of MSI was applied to 14,111 CRCs, 1756 gastric carcinomas and 506 endometrial carcinomas, which were diagnosed on the basis of current World Health Organization (WHO) classification [29] and referred for molecular analysis to the N.N. Petrov Institute of Oncology (St. Petersburg, Russia) within the years 2013–2023. The majority of the CRCs included in this study, as well as a subset of GCs, were analyzed for mutations in the *KRAS*, *NRAS* and *BRAF* oncogenes [19]. In addition, MSI testing was applied to 459 cervical carcinomas, 64 cholangiocarcinomas, 339 ovarian cancers and 474 pancreatic cancers. In the years 2013–2021, microsatellite analysis was largely based on the use of a single marker, *BAT26*, given the evidence for its high accuracy for MSI detection [30]. In the years 2022–2023, this assay was replaced by the standard MSI test involving 5 markers (*BAT25*, *BAT26*, *NR21*, *NR22* and *NR24*). MSI analysis was generally performed using tumor tissues only; in exceptionally rare instances of ambiguous results, corresponding normal cells were utilized for comparison. The primers and probes for these assays are described in Appendix A. For the pentaplex panel, tumors with two or more shifts were classified as MSI-positive [31]. Capillary electrophoresis was performed using the GenomeLab GeXP Genetic Analysis System (Beckman Coulter, Brea, CA, USA) or the Nanophore-05 instrument (Syntol, Moscow, Russia). 

The detection of *ALK*, *ROS1*, *RET* and *NTRK1–3* rearrangements was carried out for 619 microsatellite-unstable tumors. The methodology of the analysis of translocations in the above genes has been described in [15,16]. Briefly, tumor blocks were compared against corresponding histological slides, and areas with sufficient content of malignant cells were dissected from the specimens. The extraction of nucleic acids from manually dissected tumor cells involved the simultaneous isolation of both DNA and RNA, followed by cDNA synthesis. The quality of cDNA was controlled by PCR amplification of the *SDHA*-specific transcript; samples with a cycle threshold (Ct) above 35 were considered unreliable for further analysis. The *ALK*, *ROS1*, *RET*, *NTRK1*, *NTRK2* and *NTRK3* genes were subjected to tests for 5′/3′-end unbalanced expression. The primers and probes for these assays are described in Appendix A. In addition to the 5′/3′-end unbalanced expression screening test, all MSI-positive tumors were analyzed by variant-specific PCR for the most common translocations affecting the *ALK* (4 variants), *ROS1* (10 variants) and *RET* (11 variants) genes, as well as *BCR::PKHD1* and *CLIP1::LTK* gene fusions [17,18]. The design and multiplexing of the variant-specific tests are described in Supplementary Appendix A. Tumors with unbalanced 5′/3′-end *ALK*, *ROS1*, *RET*, *NTRK1*, *NTRK2* or *NTRK3* expression, which lacked the above-mentioned common rearrangements, were further tested for rare translocation variants (see the list in Supplementary Appendix A). In addition, we applied this testing for all MSI+ tumors with detectable *RET* and *NTRK1–3* expression, even in the absence of 5′/3′-end expression imbalance. Finally, samples with unbalanced expression, which did not have PCR-detectable rearrangements, were subjected to RNA-based NGS. 

PCR reactions were performed using the CFX-96 Real-Time PCR Detection System (Bio-Rad, Hercules, CA, USA). The PCR mix contained 1 μL of cDNA sample, 1× GeneAmp PCR Buffer I (Thermo Fisher Scientific Baltics UAB, Vilnius, Lithuania, cat. #4379876), 250 μM of each dNTP, 200 nM of each primer and probe, 2.5 mM MgCl_2_ and 1 U of TaqM polymerase (AlkorBio, St. Petersburg, Russia, cat. #751-100) in a total volume of 20 μL. The PCR reactions were initiated by enzyme activation (95 °C, 10 min) and included 38 cycles (95 °C for 15 s followed by 58 °C for 1 min). 

NGS RNA sequencing was performed using the custom QIAseq RNA-Scan Targeted Panel (Qiagen, Hilden, Germany, cat. #CFHS10753Z-81), which is capable of detecting rearrangements in 6 genes (*ALK*, *ROS1*, *RET*, *NTRK1*, *NTRK2* and *NTRK3*). This panel relies on the technology of Single Primer Extension (SPE). RNA was obtained using the PureLink FFPE RNA Isolation kit (Invitrogen, Carlsbad, CA, USA, cat. #45-7015). NGS was carried out on the Illumina MiSeq or NextSeq 550 instruments. Fusion calling was performed with the STAR-Fusion pipeline (V.1.4.0). 

## 5. Conclusions

Overall, this study demonstrates that the comprehensive analysis of *ALK*, *RET*, *NTRK1*, *NTRK2* and *NTRK3* rearrangements is particularly feasible in MSI-positive *KRAS/NRAS/BRAF* mutation-negative CRCs, although there is also a moderate frequency of these events in other categories of microsatellite-unstable tumors. The spectrum of the involved tyrosine kinases and their partners is characterized by a high level of diversity; therefore, the utilization of indirect methods, such as IHC or FISH, may be associated with some uncertainty. While RNA-based NGS is the gold standard for the detection of gene rearrangements, the diagnostic pipeline presented here may be considered as a cost-efficient alternative for facilities with limited access to massive parallel sequencing.

## Figures and Tables

**Figure 1 ijms-24-13610-f001:**
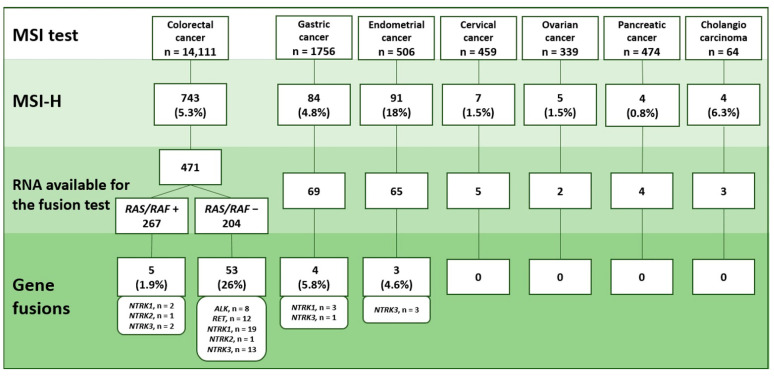
Flowchart of the analysis of gene rearrangements in MSI-positive samples. *RAS/RAF* +: CRCs carrying mutations in *KRAS*, *NRAS* or *BRAF* oncogenes. *RAS/RAF* −: *KRAS/NRAS/BRAF* wild-type CRCs.

**Figure 2 ijms-24-13610-f002:**
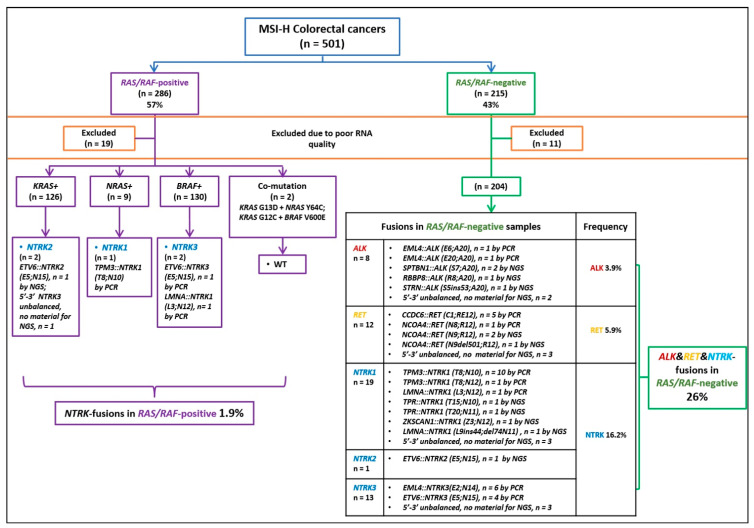
Flowchart of the detection of gene rearrangements in MSI-positive samples of colorectal cancers. *RAS/RAF*—*KRAS/NRAS/BRAF* mutation status.

**Table 1 ijms-24-13610-t001:** Clinical data for MSI-positive tumors with tyrosine kinase gene fusions.

	ID	Sex	Age, Years	Diagnosis	*RAS*/*RAF* Mutations	5′/3′-End Expression Imbalance	Fusion	Method of Fusion Identification
1.	A9365	m	74	CRC	negative	*ALK*	*STRN::ALK (S5;ins53A20)*	NGS
2.	C1182	f	71	CRC	negative	*ALK*	*EML4::ALK (E6;A20)*	PCR
3.	C3234	f	49	CRC	negative	*ALK*	*SPTBN1::ALK (S7;A20)*	NGS
4.	D0934	m	70	CRC	negative	*ALK*	*RBBP8::ALK (R8;A20)*	NGS
5.	P28580	f	66	CRC	negative	*ALK*	*SPTBN1::ALK (S7;A20)*	NGS
6.	P30811	m	80	CRC	negative	*ALK*	*EML4::ALK (E20;A20)*	PCR
7.	B9775	f	62	CRC	negative	-	*CCDC6::RET (C1;RE12)*	PCR
8.	C7622	m	58	CRC	negative	-	*CCDC6::RET (C1;RE12)*	PCR
9.	A7310	m	65	CRC	negative	-	*CCDC6::RET (C1;RE12)*	PCR
10.	P22395	m	62	CRC	negative	-	*CCDC6::RET (C1;RE12)*	PCR
11.	B8013	f	63	CRC	negative	*RET*	*NCOA4::RET (N8;R12)*	PCR
12.	C2542	f	67	CRC	negative	*RET*	*NCOA4::RET (N9del501;R12)*	NGS
13.	C4286	m	54	CRC	negative	*RET*	*NCOA4::RET (N9;R12)*	NGS
14.	P16667	f	65	CRC	negative	*RET*	*NCOA4::RET (N9;R12)*	NGS
15.	P24158	f	66	CRC	negative	*RET*	*CCDC6::RET (C1;RE12)*	PCR
16.	A4695	f	64	CRC	negative	*NTRK1*	*TPM3::NTRK1 (T8;N10)*	PCR
17.	B2240	f	60	CRC	negative	*NTRK1*	*TPM3::NTRK1 (T8;N10)*	PCR
18.	B9472	m	52	CRC	negative	*NTRK1*	*TPM3::NTRK1 (T8;N10)*	PCR
19.	C1589	f	64	CRC	negative	*NTRK1*	*TPM3::NTRK1 (T8;N10)*	PCR
20.	C6335	f	69	CRC	negative	*NTRK1*	*TPM3::NTRK1 (T8;N10)*	PCR
21.	C6755	m	72	CRC	negative	*NTRK1*	*TPM3::NTRK1 (T8;N10)*	PCR
22.	D0273	m	68	CRC	negative	*NTRK1*	*TPM3::NTRK1 (T8;N10)*	PCR
23.	D1407	f	58	CRC	negative	*NTRK1*	*TPM3::NTRK1 (T8;N10)*	PCR
24.	E2169	m	58	CRC	negative	*NTRK1*	*TPM3::NTRK1 (T8;N10)*	PCR
25.	P18891	f	59	CRC	negative	*NTRK1*	*TPM3::NTRK1 (T8;N12)*	PCR
26.	P21693	m	86	CRC	negative	*NTRK1*	*TPM3::NTRK1 (T8;N10)*	PCR
27.	E4114	f	69	CRC	negative	*NTRK1*	*LMNA::NTRK1 (L3;N10)*	PCR
28.	B9540	m	72	CRC	negative	*NTRK1*	*TPR::NTRK1 (T15;N10)*	NGS
29.	C4763	m	65	CRC	negative	*NTRK1*	*TPR::NTRK1 (T20;N11)*	NGS
30.	C7112	f	65	CRC	negative	*NTRK1*	*ZKSCAN1::NTRK1 (Z3;N12)*	NGS
31.	D0155	f	81	CRC	negative	*NTRK1*	*LMNA::NTRK1 (L9ins44;del74N11)*	NGS
32.	B9665	m	74	CRC	negative	*NTRK2*	*ETV6::NTRK2 (E5;N15)*	NGS
33.	A4890	f	65	CRC	negative	*NTRK3*	*EML4::NTRK3 (E2;N14)*	PCR
34.	B1263	f	58	CRC	negative	-	*EML4::NTRK3 (E2;N14)*	PCR
35.	B4631	f	68	CRC	negative	*NTRK3*	*EML4::NTRK3 (E2;N14)*	PCR
36.	C3968	f	70	CRC	negative	*NTRK3*	*EML4::NTRK3 (E2;N14)*	PCR
37.	C5878	m	57	CRC	negative	*NTRK3*	*ETV6::NTRK3 (E5;N15)*	PCR
38.	C6430	f	72	CRC	negative	*NTRK3*	*EML4::NTRK3 (E2;N14)*	PCR
39.	C7912	f	54	CRC	negative	*NTRK3*	*EML4::NTRK3 (E2;N14)*	PCR
40.	D7461	f	61	CRC	negative	*NTRK3*	*ETV6::NTRK3 (E5;N15)*	PCR
41.	P22904	f	68	CRC	negative	*NTRK3*	*EML4::NTRK3 (E2;N14)*	PCR
42.	P23972	f	67	CRC	negative	*NTRK3*	*ETV6::NTRK3 (E5;N15)*	PCR
43.	B4561	m	31	CRC	*KRAS* p.A146T	*NTRK2*	*ETV6::NTRK2 (E5;N15)*	NGS
44.	C1274	f	73	CRC	*BRAF* p.V600E	-	*ETV6::NTRK3 (E5;N15)*	PCR
45.	C3231	f	76	CRC	*BRAF* p.V600E	*NTRK1*	*LMNA::NTRK1 (L3;N10)*	PCR
46.	P29082	f	85	CRC	*NRAS* p.Q61K	*NTRK1*	*TPM3::NTRK1 (T8;N10)*	PCR
47.	C7527	f	71	GC	*KRAS* p.G12C	*NTRK1*	*TPM3::NTRK1 (T8;N10)*	PCR
48.	C4292	f	72	CRC	Negative	*ALK*	not defined	n/a for NGS
49.	C8576	m	66	CRC	Negative	*ALK*	not defined	n/a for NGS
50.	C2716	m	80	CRC	Negative	*RET*	not defined	n/a for NGS
51.	C5821	m	83	CRC	Negative	*RET*	not defined	n/a for NGS
52.	P15387	m	68	CRC	Negative	*RET*	not defined	n/a for NGS
53.	B9814	f	65	CRC	Negative	*NTRK1*	not defined	n/a for NGS
54.	C4831	f	60	CRC	Negative	*NTRK1*	not defined	n/a for NGS
55.	D0795	m	74	CRC	Negative	*NTRK1*	not defined	n/a for NGS
56.	C5665	f	68	CRC	Negative	*NTRK3*	not defined	n/a for NGS
57.	C9396	m	32	CRC	Negative	*NTRK3*	not defined	n/a for NGS
58.	D3390	f	80	CRC	Negative	*NTRK3*	not defined	n/a for NGS
59.	D5144	m	66	CRC	*KRAS* p.G12D	*NTRK3*	not defined	n/a for NGS
60.	A9663	m	66	GC	Negative	*NTRK1*	not defined	n/a for NGS
61.	C1642	m	83	GC	Negative	*NTRK1*	not defined	n/a for NGS
62.	E3046	m	34	GC	Negative	*NTRK3*	not defined	n/a for NGS
63.	C8708	f	67	EC	Negative	*NTRK3*	not defined	n/a for NGS
64.	D5486	f	71	EC	Negative	*NTRK3*	not defined	n/a for NGS
65.	D6096	f	64	EC	Negative	*NTRK3*	not defined	n/a for NGS

CRC—colorectal cancer, EC—endometrial cancer, GC—gastric cancer, f—female, m—male, n/a for NGS—material not available for next generation sequencing.

## Data Availability

The data that support the findings of this study are available from the corresponding author upon reasonable request.

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
