# Peer review of "ALK*, *ROS1*, *RET* and *NTRK1–3* Gene Fusions in Colorectal and Non-Colorectal Microsatellite-Unstable Cancers"

_ijms, 2023, doi:10.3390/ijms241713610_

Round 1

Reviewer 1 Report

This study demonstrates that some population of microsatellite-unstable cancers have gene rearrangements in ALK, RET and NTRK. The report indicates that gene fusions occurred in cancers with microsatellite instability and without KRAS/NRAS/BRAF mutations.

1. Table 1 may needs to be revised to add the annotation for the abbreviations. The table includes the data without tyrosine kinase gene fusions, whereas the title of the table is Clinical data for MSI-positive tumors with tyrosine kinase gene fusions. Please clarify it. It should be cited in the text near the Table1.
2. The reason why a quite few gene fusions are observed in KRAS/NRAS/BRAF wild-type colorectal cancer would be discussed in 3. Discussion.
3. Please spell out the abbreviations at the first time they appear.
4. Conclusion section may be added to summarize the results of the study.
5. Please check the whole manuscript very carefully.

Author Response

Comment: Table 1 may needs to be revised to add the annotation for the abbreviations. The table includes the data without tyrosine kinase gene fusions, whereas the title of the table is Clinical data for MSI-positive tumors with tyrosine kinase gene fusions. Please clarify it. It should be cited in the text near the Table1.

Response: Thank you very much for noticing a mistake! Some tumors mentioned in the Table 1 do contain fusions, however the translocation variants could not be identified because the samples were not available for NGS. We changed the wording and corrected all other deficiencies mentioned by the Reviewer.

Comment: The reason why a quite few gene fusions are observed in KRAS/NRAS/BRAF wild-type colorectal cancer would be discussed in the Discussion.

Response: We now comment on this issue: “Our report confirms that alterations in genes belonging to MAPK pathway are generally mutually exclusive. Indeed, activation of a single member of this molecular cascade, be it receptor tyrosine kinase, or KRAS, NRAS or BRAF oncogene, is usually sufficient to drive the signaling. Furthermore, mentioned above genetic alterations are usually considered to be more or less equivalent in terms of phenotypic consequences [7].”

Comment: Please spell out the abbreviations at the first time they appear.

Response: This is done.

Comment: Conclusion section may be added to summarize the results of the study.

Response: The Conclusion section has been added.

Comment: Please check the whole manuscript very carefully.

Response: We have subjected the paper to the additional round of examination.

Reviewer 2 Report

This research analyzed microsatellite instability in a large series of cases, and also analyzed gene fusion of other genes. Data showed that in KNB-negative cases (KRAS/NRAS/BRAF wild-type colorectal cases), the number of gene fusions was higher. The manuscript is well written, and it is easy to read, and understand. The manuscript would improve if more clinical data was available, such as survival analysis, but this data may not be available. Also, I wonder if these NKB mutation and gene fusions would be identified in MSI negative cases.

Specific comments:

(1) For the detection of MSI, a single marker was used during the period 2013-2023 (BAT26). But in the period 2023-2023 the method was replaced by using 5 markers (BAT25,BAT26, NR21, NR22 and NR24). Could you please explain how comparable are the two methods? Do they have same sensitivity? Could a change of method create a bias? What percentage of cases were analyzed for each method?

(2) Line 279. Could you please explain how the tumors cells were manually dissected? “Crude” microdissection?

(3) Could you please add the catalog number of the reagents that were used (where appropriate)?

(4) Lines 262-263. Were the neoplastic cases diagnosed based on the current WHO classification?

(5) In the analysis using BAT26, was the tumor compared with normal mucosa or normal tissue of the same patient?

(6) If you look at the dMMR data of the TCGA, are your percentages of cases with MSI-High comparable/similar?

(7) Was the immune checkpoint analyzed in the positive cases? Did any patient receive immune checkpoint inhibitor immunotherapy?

(8) Line 81. How was the RNA quality control tested?

(9) Line 90. How was the NGS performed? Is the variant-specific PCR comparable to the NGS?

(10) In Table 1 there are 65 cases described, but in line 85 says 59 tumors. Is this correct? From cases 37 to 65, there is no information of the fusion partner, or is negative. Is this correct?

(11) In Figure 1, it is written “KNB?”. Should not be “KNB-”? (In Figure 2, it is written “NKB-negative”).

(12) Line 177. Why is the cutoff of 50 years biological important?

(13) In Figure 2. Is the 26% vs 19% statistically significant? If you look at MSI negative cases, would you expect a similar number of fusion percentages?

(14) Apart from the analyzed genes, do MSI-High also affect other relevant genes and/or parthways?

Author Response

Comment: The manuscript would improve if more clinical data was available, such as survival analysis, but this data may not be available. Also, I wonder if these KNB mutation and gene fusions would be identified in MSI negative cases.

Response: We now mention in the Discussion: “This study has some limitations. In particular, we did not have a survival data for the patients, therefore, we could not evaluate whether the presence of gene rearrangements affects the disease outcomes.” Furthermore, we incorporated more details regarding the frequencies of various events: “Indeed, activation of a single member of this molecular cascade, be it receptor tyrosine kinase, or KRAS, NRAS or BRAF oncogene, is usually sufficient to drive the signaling. Furthermore, mentioned above genetic alterations are usually considered to be more or less equivalent in terms of phenotypic consequences [7]. Interestingly, despite this apparent equivalence, the frequencies of alterations of particular genes vary considerably between MSI-positive and MSI-negative CRCs. Microsatellite unstable CRCs have about twice lower frequency of KRAS mutations but approximately 4 times higher incidence of BRAF mutations as compared to MSI-negative tumors [7,19]. Furthermore, while kinase gene rearrangements are common in CRCs with MSI, they are exceptionally rare in MSI-negative colorectal carcinomas [11]. Overall, the cumulative frequency of the activation of MAPK cascade is similar in CRCs with and without MSI: for example, 320/501 (63.9%) microsatellite unstable CRCs analyzed within this study had evidence for genetic alteration of the MAPK pathway (Figure 2), which is nearly identical to the estimates obtained for microsatellite stable colorectal carcinomas [19].” 

Comment: For the detection of MSI, a single marker was used during the period 2013-2023 (BAT26). But in the period 2023-2023 the method was replaced by using 5 markers (BAT25,BAT26, NR21, NR22 and NR24). Could you please explain how comparable are the two methods? Do they have same sensitivity? Could a change of method create a bias? What percentage of cases were analyzed for each method?

Response: We focus on this analysis in the Section 2.1: “Since we utilized both single-marker BAT26 testing and the pentaplex panel (BAT25, BAT26, NR21, NR22 and NR24), we evaluated whether the differences between these two methods are likely to influence the results. We considered the BAT26 length in 471 CRCs, 69 GCs and 66 ECs, which were microsatellite unstable by the pentaplex assay. Strikingly, only 3 out of 586 these tumors had normal BAT26 status indicating that the methodology of MSI testing had no significant influence on the results.”

Comment: Line 279. Could you please explain how the tumors cells were manually dissected? “Crude” microdissection?

Response: We now describe the procedure in the Material and Methods: “Briefly, tumor blocks were compared against corresponding histological slides, and areas with sufficient content of malignant cells were dissected from the specimens.”

Comment: Could you please add the catalog number of the reagents that were used (where appropriate)?

Response: The catalog numbers have been inserted.

Comment: Lines 262-263. Were the neoplastic cases diagnosed based on the current WHO classification?

Response: We now mention that the carcinomas “…were diagnosed on the basis of current World Health Organization (WHO) classification [29]”.

Comment: In the analysis using BAT26, was the tumor compared with normal mucosa or normal tissue of the same patient?

Response: We now mention in the Material and Methods:  “MSI analysis was generally performed using tumor tissues only; in exceptionally rare instances of ambiguous results corresponding normal cells were utilized for the comparison.”

Comment: If you look at the dMMR data of the TCGA, are your percentages of cases with MSI-High comparable/similar?

Response: We now state that “These [MSI] frequencies are in good agreement with the published data [1].”

Comment: Was the immune checkpoint analyzed in the positive cases? Did any patient receive immune checkpoint inhibitor immunotherapy?

Response: The analysis of treatment data is beyond the scope of this manuscript.

Comment: Line 81. How was the RNA quality control tested?

Response: We now state that The quality of cDNA was controlled by PCR amplification of the SDHA-specific transcript; samples with a cycle threshold (Ct) above 35 were considered unreliable for further analysis.

Comment: Line 90. How was the NGS performed? Is the variant-specific PCR comparable to the NGS?

Response: We specify that “NGS RNA sequencing was performed using the custom QIAseq RNA-Scan Targeted Panel (Qiagen, Hilden, Germany, cat #CFHS10753Z-81), which is capable of detecting rearrangements in 6 genes (ALK, ROS1, RET, NTRK1, NTRK2, NTRK3). This panel relies on the technology of Single Primer Extension (SPE)”. We did not confirm NGS results by variant-specific PCR.

Comment: In Table 1 there are 65 cases described, but in line 85 says 59 tumors. Is this correct? From cases 37 to 65, there is no information of the fusion partner, or is negative. Is this correct?

Response: Thank you for noticing this inconsistency! The line 85 mentioned 59 tumors with 5’/3’-end expression imbalance. In addition, there were 6 tumors with fusions, which were not detected by the expression test, but we identified by variant-specific PCR. We have added these details to the Table 1.

Comment: In Figure 1, it is written “KNB?”. Should not be “KNB-”? (In Figure 2, it is written “NKB-negative”).

Response: We apologize, this is a mistyping. We have corrected it.

Comment:  Line 177. Why is the cutoff of 50 years biological important?

Response: We mention that “This age threshold is commonly utilized for discrimination between sporadic CRCs, which developed microsatellite instability due to hypermethylation of MLH1 promoter, and hereditary CRCs, which is attributed to the biallelic inactivation of DNA mismatch repair genes [3,4,7].”

Comment: In Figure 2. Is the 26% vs 19% statistically significant? If you look at MSI negative cases, would you expect a similar number of fusion percentages?

Response: It is not 19%, it is 1.9%. This difference is statistically significant (P < 0.001).

Comment: Apart from the analyzed genes, do MSI-High also affect other relevant genes and/or parthways?

Response: We now state in the Discussion: “Overall, the cumulative frequency of the activation of MAPK cascade is similar in CRCs with and without MSI: for example, 320/501 (63.9%) microsatellite unstable CRCs analyzed within this study had evidence for genetic alteration of the MAPK pathway (Figure 2), which is nearly identical to the estimates obtained for microsatellite stable colorectal carcinomas [19]. The remaining 30-40% CRCs do not have overt genetic alterations within this signaling cascade and deserve further investigations [7].”